# MVI2V: HUMAN CENTRIC IMAGE TO VIDEO GENERATION WITH MULIVIEW CONSISTENT APPEARANCE

## ABSTRACT

Recent advances in image-to-video (I2V) generation have enabled the synthesis of high-quality videos from a single input image. However, a significant limitation emerges with this single-image conditioning: models struggle to maintain appearance consistency for unseen regions, like the back of a garment. Because they lack complete information, the models are forced to hallucinate these missing views, a process that frequently introduces visual artifacts and inconsistencies. Consequently, this issue hinders their adoption in applications such as e-commerce, where visual fidelity to the actual garment is critical. In this paper, we propose the Multiview Enhanced Image-to-Video Generation Model (MVI2V), which solves this issue by introducing other multi-views of a person or garments as extra reference images to enhance the generation process. Specifically, MVI2V introduces an additional, structurally identical forward stream dedicated to processing the reference images. This transforms the original single- or dual-stream architecture into a dual- or triple-stream one, respectively. Cross-stream fusion is facilitated by the self-attention mechanism, which enables bidirectional information flow among tokens of different types. Regarding the training strategy, we incorporate an in-painting sub-task that randomly masks the region of the person in the conditioning image, thereby compelling the model to rely more heavily on guidance from the reference images. To facilitate efficient model learning, we have meticulously designed a data curation pipeline. This pipeline selectively filters videos exhibiting large-angle variations of the subject, and then systematically extracts a comprehensive set of multi-view reference images for each video. Extensive experiments on both Wan2.1 I2V and our in-house I2V model show that our MVI2V model can accurately reference multi-views images of a person or garment images, while simultaneously preserving the foundational I2V generation capabilities of the original model, and validate the effectiveness of the proposed network architecture, training strategy, and dataset curation pipeline. Code will be released to advance the related research.

## 1 INTRODUCTION

Recently, significant progress in denoising diffusion models(DDM) has greatly enhanced the generation ability of videos from text descriptions (Yang et al., 2024; Polyak et al., 2024; Kong et al., 2024). Among the diverse applications of diffusion models, Image-to-Video (I2V) generation has emerged as a particularly promising and actively researched direction. This task aims to animate a static image based on user-provided prompts, offering an effective balance between controllability and creative flexibility. As such, it has found widespread use in entertainment, social media, and personalized content creation.

In this paper, we focus on the human-centric application of I2V, which takes a human image as input and animates the person according to user-provided text prompts. Unfortunately, in demanding scenarios such as e-commerce, where visual fidelity and appearance consistency with actual subjects or garments are critical, it is insufficient for the model to generate videos with only one conditioning image. The model is forced to hallucinate the appearance of unseen regions of the subject and garment from the first image. A straightforward solution to this is to incorporate additional input views into the model.

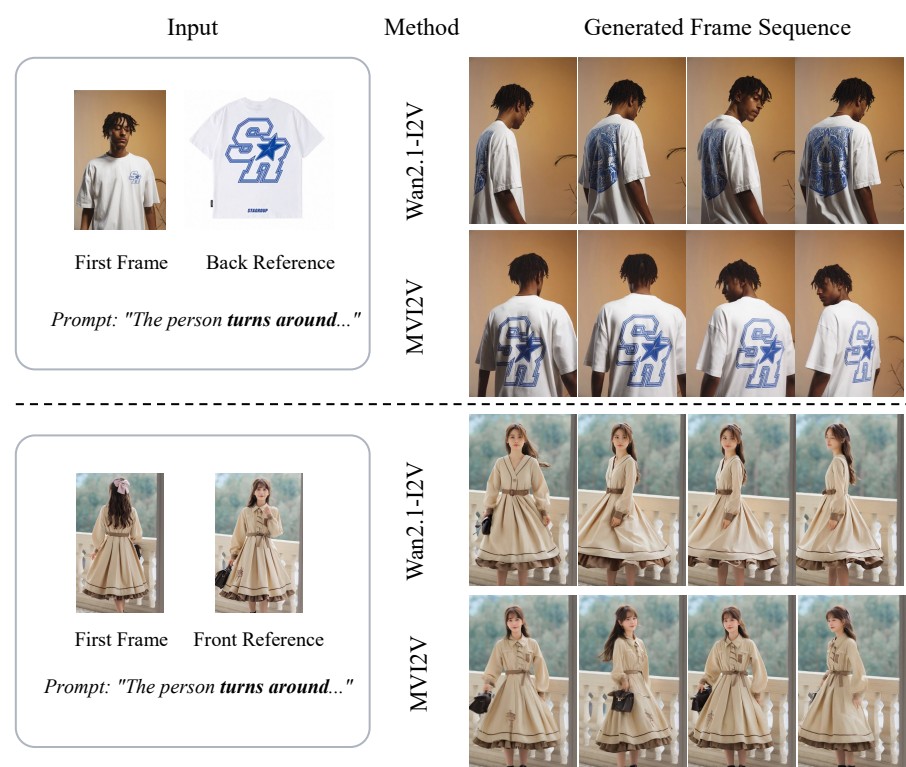

Figure 1: Our MVI2V multi-view generation ressults compared with Wan2.1-I2V baseline.

However, it is non-trivial to achieve the desired functionality from scratch. The primary challenge is the scarcity of human-centric videos featuring dynamic body reorientations, as such data occupies the long tail of the overall distribution. Such limited data makes it highly challenging to train a generalizable multi-view image-to-video generation model from scratch. Therefore, we opt to extend existing powerful image-to-video generation models and finetune them on a targeted multi-view dataset. Specifically, MVI2V extends the original model with an additional forward stream, structurally identical to the primary one for video tokens, to process multi-view reference images and enable bidirectional information flow between reference image tokens and video tokens via self-attention in each transformer block. In this way, video tokens can gather information about the unseen garment appearance from the multi-view image tokens. We named this architecture as Multiview Enhanced Image-to-Video Generation Model (MVI2V). MVI2V is general and robust, readily adaptable to existing popular video diffusion backbones, including single-stream based models (Yang et al., 2024; Kong et al., 2024) and dual-stream based models (Polyak et al., 2024; Wang et al., 2025).

We also constructed a specialized data pipeline to filter out multi-view targeted dataset to accelerate model training. Starting from a large video pool, this pipeline sequentially filters for single-person videos and then for those featuring large angles of subject rotation. Furthermore, we formulated a multi-view reference frame extraction strategy. It identifies five comprehensive views, including frontal and back perspectives. Concurrently, the top and bottom garments are segmented from the frontal and back views to augment the dataset of garment reference images. We found that this data curation pipeline substantially improves both the model's learning efficiency and its final reference capability.

Despite employing the MVI2V model and the meticulously curated dataset, we observe that the model can still "learn a shortcut" during training: it can generate a temporally consistent video while completely ignoring the reference frames, although the resulting appearance will not match them. This is because that for a pre-trained I2V model, the conditioning first frame already provides a strong prior for the subject's appearance. This tendency impedes the model's ability to learn from the new reference images. To counteract this, we introduce an auxiliary inpainting subtask.

By randomly masking out the person's region in the conditioning first frame, this strategy compels the model to reconstruct the subject's appearance by leveraging the information from soely reference images. Our experiments demonstrate that this simple approach further enhances the model's capability to adhere to the reference images.

In summary, our contributions are outlined below:

- We propose the first training framework that integrates the MVI2V architecture and an inpainting sub-task to augment existing image-to-video generation models with multi-view reference support. This framework is readily adaptable to both single-stream and dual-stream-based backbones.
- Our carefully designed dataset collection pipeline is effective to enhance the training efficency for this specialized task.
- Experiments ablates the effectiveness of our proposed MVI2V model, inpainting subtask and dataset curation pipeline.

## 2 RELATED WORK

### 2.1 VIDEO GENERATION MODELS

The success of Diffusion Models in image generation (Rombach et al., 2021; Ho et al., 2020) has spurred their exploration in the video domain (Lin et al., 2024a; Yang et al., 2024; Kong et al., 2024; Wang et al., 2025; Gao et al., 2025). Video Diffusion Models (VDMs) was the first work among them to achieve this by extending the 2D U-Net architectures of image generation models into 3D U-Net for video synthesis. Other works (Zhou et al., 2022; Gong et al., 2024; Wang et al., 2023) introduce 1D temporal attention to reduce computation overhead. The advent of the Diffusion Transformer (DiT, (Peebles & Xie, 2022)) introduced the Transformer architecture (Vaswani et al., 2017) into diffusion models, leading to substantial improvements in generation quality. Consequently, modern video generation models now widely employ the DiT architecture for modeling both visual appearance and temporal dynamics.

Modern powerful DiT-based open-source or commercial video generation models typically process multi-modal inputs, such as video and text, in two primary ways. The first approach, which we refer to as **single-stream**, models video tokens through a single set of parameters and interacts with text tokens through cross attention mechanism (*e.g.* MovieGen (Polyak et al., 2024) and Wan2.1 (Wang et al., 2025)). The second is a **dual-stream** approach based on the Multimodal Diffusion Transformer (MMDiT), which processes video and text with separate parameters, and merges them during self-attention mechanism (*e.g.* CogvideoX (Yang et al., 2024), HunyuanVideo (Kong et al., 2024) and SeeDance (Gao et al., 2025)). In practice, both methods achieve excellent results in video-text alignment and generation realism.

### 2.2 IMAGE-TO-VIDEO GENERATION

Most existing image-to-video (I2V) methods adapt pre-trained text-to-video (T2V) models to accept both image and text inputs, typically by fine-tuning on image-video pairs to inherit original model's generative ability and accelerate convergence. A common technique is to concatenate the initial frame's latent and mask with the noisy video latent in the channel dimension (Kong et al., 2024; Wang et al., 2025; Gao et al., 2025; Yang et al., 2024). Wan2.1 (Wang et al., 2025) further enhances this by injecting global semantics from the frame's CLIP embedding via cross-attention. Despite these advances, all such single-frame conditioned methods fail to guarantee multi-view appearance consistency with actual garments. Our MVI2V overcomes this limitation by introducing a dedicated forward stream for processing multiple reference images, thereby enabling the generation of videos consistent with input multi-view reference images.

## 3 PRELIMINARY

Flow matching models (Liu et al., 2022; Lipman et al., 2022) provide a theoretically grounded framework for learning continuous-time generative processes in diffusion models, and have demon-

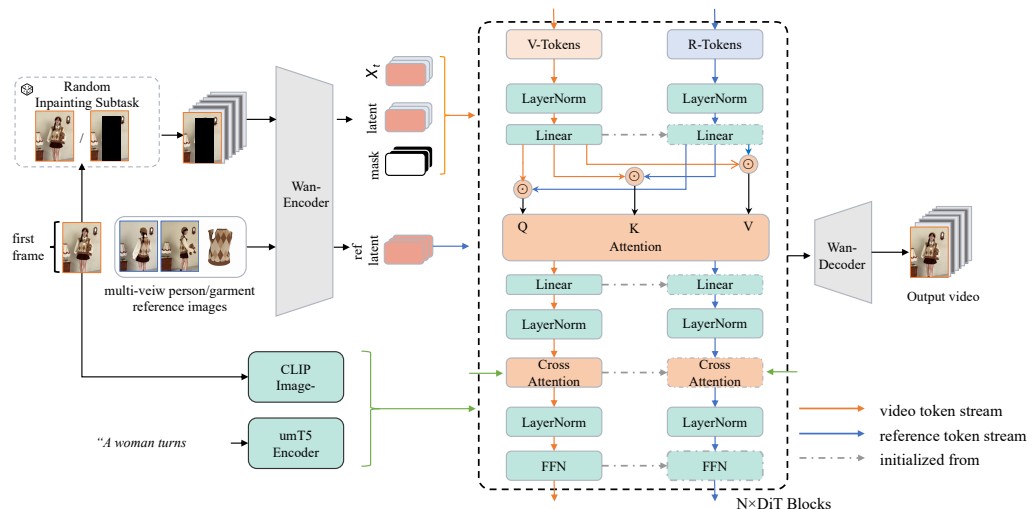

Figure 2: Multiview enhanced image-to-video generation training framework.

strated strong capabilities in visual generation. It circumvents iterative velocity prediction, enabling stable training via ordinary differential equations (ODEs) while maintaining equivalence to maximum likelihood objectives. In general practice, the probability path usually exists in a latent space, not in a pixel space, to reduce computation and speed up training. During training, the video tensor is first compressed by a video encoder $\mathcal{E}$ into latent space as $x_1$. Given a randomly sampled noise $x_0 \sim \mathcal{N}(0, \mathcal{I})$ and a timestep $t \in [0, 1]$, an intermediate latent $x_t = tx_1 + (1 - t)x_0$ is obtained as the training input. The ground truth $v_t$ is $v_t = \frac{dx_t}{dt} = x_1 - x_0$.

For image-to-video generation, the model is trained to predict the velocity, given the input prompt $c_{txt}$ and the first frame $c_{img}$. Therefore, the loss function is formulated as the mean squared error between the model output and $v_t$,

$$L = E_{x_0, x_1, c_{txt}, c_{img}, t} ||u(x_t, c_{txt}, c_{img}, t; \theta) - v_t||^2$$

In this paper, we extend the task to multiview enhanced image-to-video generation. Therefore, the model is conditioned on extra reference images $c_{ref}$ and the training loss is reformulated as:

$$L = E_{x_0, x_1, c_{txt}, c_{img}, c_{ref}, t} ||u(x_t, c_{txt}, c_{img}, c_{ref}, t; \theta) - v_t||^2$$

## 4 METHODOLOGY

### 4.1 METHOD OVERVIEW

MVI2V enables video generation from multi-view references by enhancing the powerful, pre-existing I2V models with an extra forward stream specifically for processing reference images, which is compatible with dominant video diffusion model backbones such as single-stream and dual-stream structures. To support this task, we also developed a data construction pipeline to curate a high-quality, training-efficient dataset of multi-view video-image pairs. In the following part of this section, we first introduce the training details including network architecture and inpainting subtask in Sec. 4.2. Then, we describe the dataset construction pipeline in Sec. 4.3.

### 4.2 MULTIVIEW ENHANCED IMAGE-TO-VIDEO GENERATION

To ensure multi-view consistent garment appearance in the generated videos, we condition our model on a new type of input: clean multi-view reference images. These images are fundamentally different from the other two inputs in distinct ways. Unlike semantic text prompts, they provide direct visual information. For noisy video tokens, they are linear interpolations of noise and clean

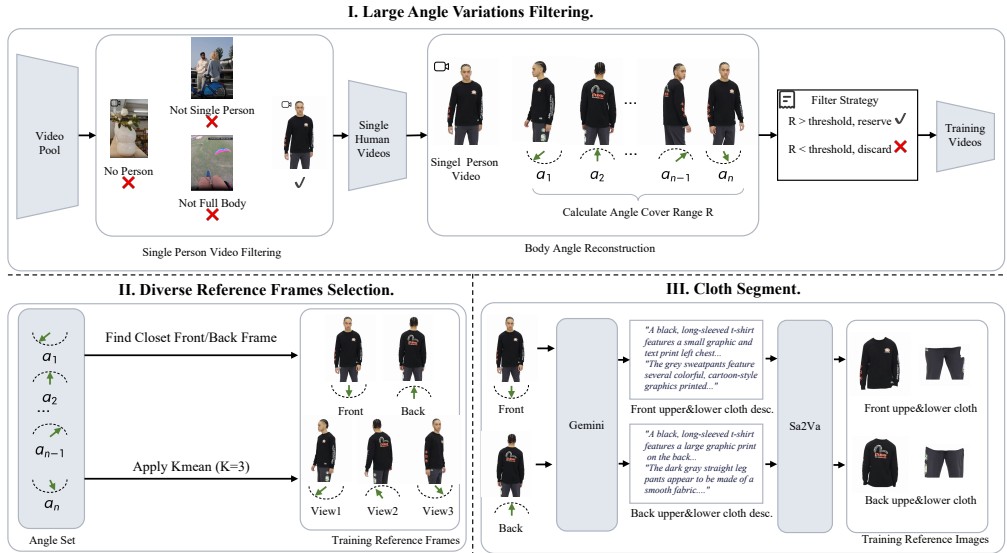

Figure 3: Our targeted data collection pipeline.

video latent, represent intermediate points along the flow-matching trajectory, and thus contain a mix of signal and noise. In contrast, clean reference frames are more similar to the endpoint of this trajectory, embodying purely clear visual information. Therefore, we treat these reference images as a unique modality in this paper.

We seek a general way to incorporate the new modality and facilitate information exchange with the other two modalities. Following MMDiT, we introduce a dedicated forward stream to process the reference images and employ self-attention to fuse its features with those from other modalities. In the following, we illustrate the architectural modifications based on Wan2.1 I2V backbone as a concrete example, which is shown in Figure 2. We leave the modification details on our in-house I2V backbone (dual stream) in Appendix B.

**MVI2V Network Architecture.** A typical transformer block of Wan2.1 consists of 3 primary modulated components: self-attention, cross-attention, and feedforward network (FFN). The forward process can be formulated as:

$$x_1 = \textbf{SelfAttention}(x; M_S, W_S)$$
$$x_2 = \textbf{CrossAttention}(x_1, c_{txt} \oplus c_{img}; M_C, W_C)$$
$$x_3 = \textbf{FFN}(x_2; M_F, W_F)$$

where $M_S, M_C, M_F$ is the collection of modulation parameters, including scale, shift, gate for self-attention, cross-attention, and feedforward layers, respectively. $W_S$ and $W_C$ are the collections of linear projection parameters to project tokens into query, key, value, and output in self-attention and cross-attention operations. And $W_F$ is the parameters of FFN. $x$ is the input video tokens, and $c_{txt}, c_{img}$ are the conditional text embeddings and CLIP embeddings of the first image. In this formulation, we omit the internal residual connections for simplicity.

As shown in the DiT block part of Fig 2, we extend each operation with an extra forward stream and parameters for the additional reference image tokens. These two modalities only interact at self-attention operation. This modified architecture is formulated as follows:

$$x_1, y_1 = \textbf{SelfAttention}(x \oplus c_{ref}; M_S, M_S', W_S, W_S')$$
$$x_2 = \textbf{CrossAttention}(x_1, c_{txt} \oplus c_{img}; M_C, W_C)$$
$$y_2 = \textbf{CrossAttention}(y_1, c_{txt} \oplus c_{img}; M_C', W_C')$$
$$x_3 = \textbf{FFN}(x_2; M_F, W_F)$$
$$y_3 = \textbf{FFN}(y_2; M_F', W_F')$$

where $M'_S, M'_C, M'_F, W'_S, W'_C, W'_F$ are the additional parameters to process multi-view reference images specifically.

To accelerate convergence, a straightforward approach is to initialize the additional parameters by duplicating the weights of the pretrained one. However, this method doubles the number of trainable parameters, significantly increasing computational overhead and memory footprint. Therefore, we propose a more parameter-efficient strategy employing Low-Rank Adaptation (LoRA) (Hu et al., 2021). Instead of full parameter replication, we augment each original weight matrix with low-rank matrices as follows:

$$W'_S = W_S + \alpha A_S B_S$$
$$W'_C = W_C + \alpha A_C B_C$$
$$W'_F = W_F + \alpha A_F B_F$$

where $\alpha$ is the scaling factor for the LoRA weights. The matrices $A_{(.)}$ and $B_{(.)}$ are the down-projection and up-projection matrices, respectively, for the LoRA layers applied to different components of the model. This parameter-efficient approach significantly reduces the number of trainable parameters.

**Inpainting Subtask.** Extending a pretrained Image-to-Video (I2V) model to incorporate multi-view references poses a significant challenge: the conditioning first frame already provides a strong appearance prior. This strong prior may cause the model to simply disregard the multi-view reference images and generate the video based solely on the conditioning frame. This presents a data construction challenge, as reference images must provide novel viewpoints distinct from the first frame to encourage the model to learn complementary features. This necessity inherently limits the range of applicable training data.

To address this and improve data utilization, we propose a simple yet effective strategy: randomly masking the human region in the conditioning frame during training. This forces the model to learn the subject's appearance exclusively from the reference images, thereby decoupling it from the conditioning frame's identity. Experiments validate that this approach significantly enhances the model's ability to synthesize subjects based on reference images.

### 4.3 DATASET CONSTRUCTION

Figure 3 shows the overall data collection pipeline. We begin by filtering a large video pool using a human detection model (Yang et al., 2023). We discard videos that contain no people, multiple people, or only part of one person, retaining only single-person videos with a sufficiently large subject.

**Large Angle Variations Filtering.** For each left video clip, we estimate the per-frame body orientation angles using a human reconstruction model (Kanazawa et al., 2018). Clips with an angular cover range greater than a predefined threshold are selected for training, while others are filtered out.

**Diverse Reference Frames Selection.** Our reference frame selection is a two-step procedure. First, recognizing that frontal and back views provide the most comprehensive information on clothing appearance, we identify the two frames closest to these canonical views. Second, to ensure viewpoint diversity, we apply k-means clustering (k=3) to all orientation angles in the clip and select the three frames closest to the resulting cluster centroids.

**Cloth Segment.** Finally, to address the data scarcity of the flattened map, we employ the gemini (Team et al., 2024) and Sa2va (Yuan et al., 2025) model to segment the upper and lower garments from the frontal and back reference frames, respectively. These segmented regions are then incorporated as additional reference frames for training. In the final dataset, each video is associated with nine reference images: two frontal/back views, three clustered multi-view images, and four segmented garment images.

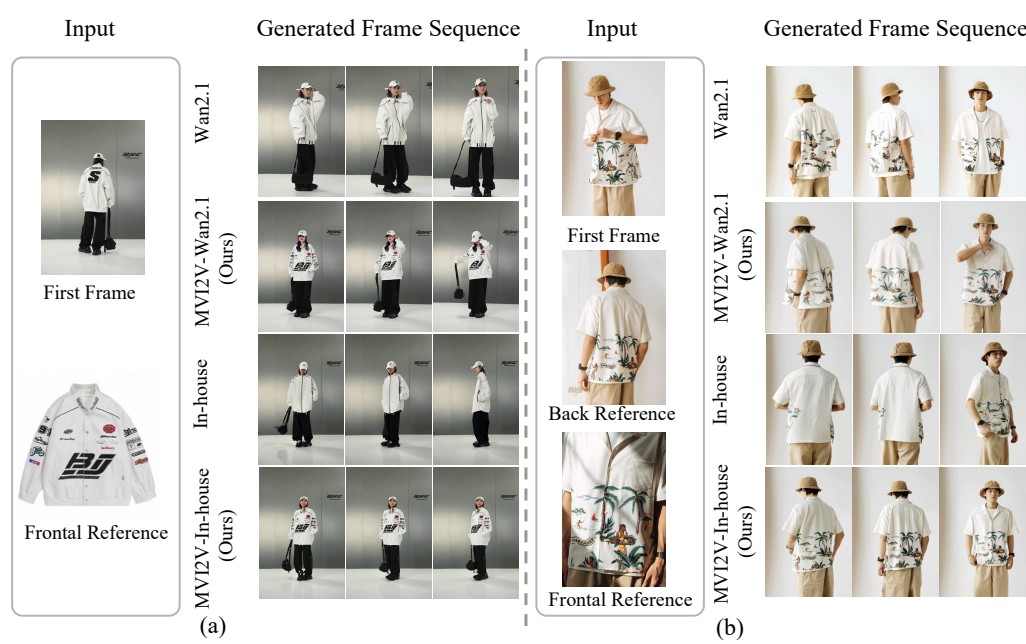

Figure 4: Qualitative comparisons with conventional image-to-video baselines using garment (a) or person (b) images as reference.

## 5 EXPERIMENTS

### 5.1 BASELINES AND METRICS

**Baselines.** To comprehensively validate the effectiveness of our proposed architectural modifications, we conduct experiments on both single-stream and dual-stream baselines. For the single-stream model, we adopt Wan2.1 (I2V-14B-720P) as the baseline. For the dual-stream model, we utilize an in-house 15B image-to-video generation model as the baseline. This model features a hybrid dual-stream/single-stream architecture, similar to Flux (Labs, 2024). We extend their single-stream blocks and dual-stream blocks into dual-stream blocks and triplet-stream blocks according to Sec 4, resulting in our MVI2V-Wan2.1 and MVI2V-In-House model, respectively. Furthermore, an naive approach to incorporate multi-view reference images is to directly concatenate reference tokens with the video tokens. The combined sequence is then processed through Transformer blocks, after which only the video tokens are isolated for decoding. We term this method Token-Concat. To compare this naive structure against our MVI2V architecture, we implemented Token-Concat on the Wan2.1 model and compare it with our MVI2V-Wan2.1 model.

To evaluate multi-view reference capability, we construct a new benchmark tailored for this task. Specifically, we manually collected multi-view images of the same subject from e-commerce websites. For each set of images, one view serves as the conditioning first frame, while the others serve as reference images. Our benchmark comprises 130 instances with multi-view person images as reference, and 85 instances with garment images as reference, totaling 205 test samples. The evaluation metrics are categorized into the following three types:

**Garment Consistency.** We assess garment consistency between the generated video frames and the reference images using GPT Achiam et al. (2023) and DINO (Caron et al., 2021). For GPT-based metrics, inspired by VQAScore (Lin et al., 2024b), we provide detailed instructions and rubrics to guide the GPT to grade the generated videos across garment consistency dimension. For the DINO-based metric, we compute the cosine similarity of DINO features between each generated frame and reference images as the garment consistency score. We apply the two scoring strategies to the multi-view person and garment evaluation sets, respectively, yields four distinct metrics $GPT_{person}, GPT_{garment}, DINO_{person}, DINO_{garment}$.

Table 1: Quantitative comparison with image-to-video baselines on our multi-view benchmark.

| Model | Garment Consistency | | | | Video Quality | | | Text-Motion Alignment |
|---|---|---|---|---|---|---|---|---|
| | $GPT_{person}$ | $GPT_{garment}$ | $DINO_{person}$ | $DINO_{garment}$ | Subject Consistency | Background Consistency | Motion Smoothness | GPT |
| Wan2.1 | 0.752 | 0.409 | 0.667 | 0.582 | 0.916 | 0.928 | 0.986 | 0.812 |
| MVI2V-Wan2.1 | **0.862** | **0.585** | **0.728** | **0.656** | **0.927** | **0.938** | **0.988** | **0.879** |
| In-House | 0.692 | 0.416 | 0.700 | 0.601 | 0.926 | 0.934 | **0.993** | **0.935** |
| MVI2V-In-House | **0.868** | **0.607** | **0.776** | **0.633** | **0.930** | **0.934** | **0.993** | 0.880 |

Table 2: Ablation results. The '+' symbol denotes the addition of a core component to the configuration of the preceding row.

| Model | Ablation Item | | | Garment Consistency | | | | Video Quality | | | Text-Motion Alignment |
|---|---|---|---|---|---|---|---|---|---|---|---|
| | MVI2V Architecture | Dataset Strategy | Inpainting Subtask | $GPT_{person}$ | $GPT_{garment}$ | $DINO_{person}$ | $DINO_{garment}$ | Subject Consistency | Background Consistency | Motion Smoothness | GPT |
| Wan2.1 | ✗ | ✗ | ✗ | 0.752 | 0.409 | 0.667 | 0.582 | 0.916 | 0.928 | 0.986 | 0.812 |
| + MVI2V Architecture | ✓ | ✗ | ✗ | 0.784 | 0.470 | **0.744** | 0.631 | 0.905 | 0.923 | 0.982 | 0.787 |
| + Dataset Strategy | ✓ | ✓ | ✗ | 0.853 | 0.562 | 0.743 | 0.638 | 0.916 | 0.932 | 0.986 | 0.857 |
| + Inpainting Subtask (Ours) | ✓ | ✓ | ✓ | **0.862** | **0.585** | 0.728 | **0.656** | **0.927** | **0.938** | **0.988** | **0.879** |
| Token-Concat | ✗ | ✓ | ✓ | 0.797 | 0.512 | 0.715 | 0.620 | 0.905 | 0.923 | 0.982 | 0.867 |

**Video Quality.** Consistent with previous work, we evaluate the "Subject Consistency", "Background Consistency", and "Motion Smoothness" metrics in VBench (Zheng et al., 2025) to conduct how the two major challenges – temporal consistency and smoothness – are addressed.

**Text-Motion Alignment.** We also use GPT based VQAScore to quantify the alignment of subject's motion with the input prompt.

## 5.2 QUALITATIVE AND QUANTITATIVE EVALUATION

Figure 4 presents a qualitative comparison between our MVI2V variants and their I2V baseline. When the subject's pose aligns with the viewpoint of the reference image, our method successfully leverages its texture details to reconstruct an appearance consistent with the actual garment. In contrast, the baseline methods either generate simplistic solid colors or inpaint incorrect, hallucinated details. We provide more qualitative comparisons in Appendix F.

The quantitative results, presented in Table 1, are consistent with our qualitative findings. Our method demonstrates significant improvements over the baseline in garment consistency, as measured by both GPT and DINO based metrics. Furthermore, the high-quality training data, produced by our carefully designed collection pipeline, also leads to notable gains in other metrics on video quality and text-motion alignment dimensions.

## 5.3 ABLATION STUDY

We evaluate the incremental addition of the three core components of MVI2V: (1) the MVI2V architecture, (2) our data collection strategy, and (3) the inpainting subtask, building upon the Wan2.1 baseline. The quantitative results are shown in Tab. 2. First, integrating the MVI2V architecture alone improves garment consistency but slightly compromises video quality and text-motion alignment. Once our data strategy and inpainting task are also

Table 3: Quantitative results on the original VBench I2V benchmark.

| Model | Subject Consistency | Background Consistency | Motion Smoothness |
|---|---|---|---|
| Wan2.1 | 0.938 | 0.969 | 0.980 |
| MVI2V-Wan2.1 | 0.953 | 0.969 | 0.987 |
| In-House | 0.965 | 0.964 | 0.994 |
| MVI2V-In-House | 0.959 | 0.958 | 0.993 |

included, almost all metrics steadily increase. Moreover, replacing the MVI2V architecture with Token-Concat leads to a significant performance drop, which underscores the superiority of our network design in handling multi-view image inputs.

Figure 5 illustrates the qualitative results of our ablation study. It is observed that, compared to the baseline, employing MVI2V architecture alone is insufficient for the model to adhere to reference images. With the addition of our proposed data collection strategy, the model gains some reference

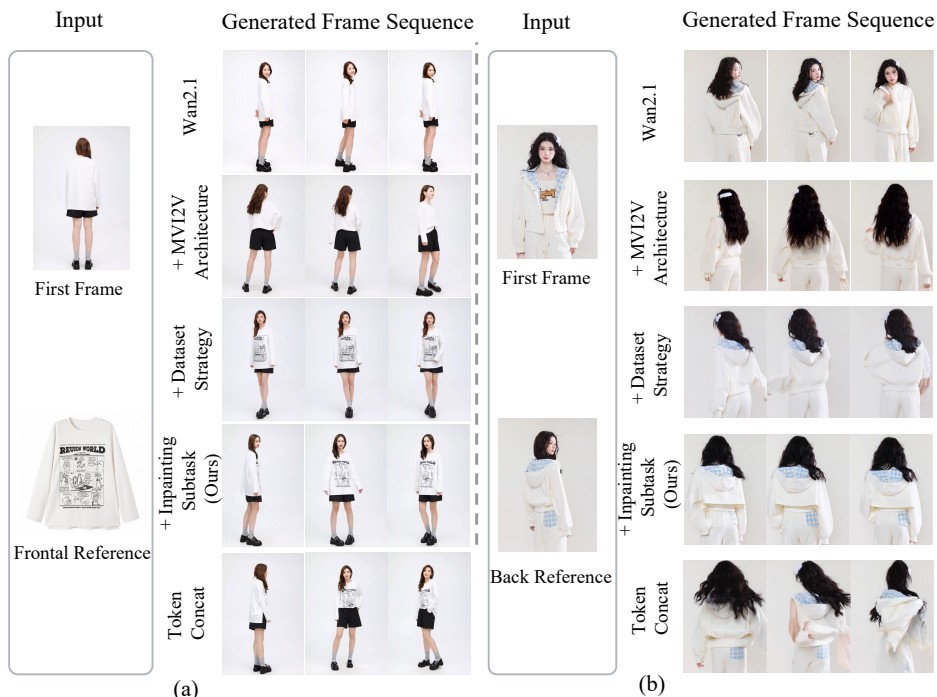

Figure 5: Qualitative ablation results using garment (a) or person (b) images as reference. Our method behaves better on texture inconsistencies (a) or loss (b) problems.

capability, but still suffer from texture inconsistencies or loss. Finally, by incorporating the inpainting sub-task, the model's referencing ability is maximized, successfully generating videos where the garment appearance is consistent with the reference image. Alternatively, replacing MVI2V architecture with the Token-Concat architecture grants some referencing ability, but introduces incorrect clothing geometry and noticeable motion artifacts due to the mixed modeling of the two modalities.

### 5.4 PRESERVATION OF FOUNDATIONAL I2V CAPABILITIES

Our MVI2V models can also function as an I2V model, retaining the capability for single-image to video generation. To validate this, we evaluated their I2V performance on the original Vbench benchmark. Table 3 compares the Vbench metrics of the MVI2V models with its baseline version. The extended models show comparable performance across all metrics. This indicates that our MVI2V training process does not compromise the original I2V capabilities. Figure 6 shows one visualization of single-image to video generation result of our MVI2V-Wan2.1 model.

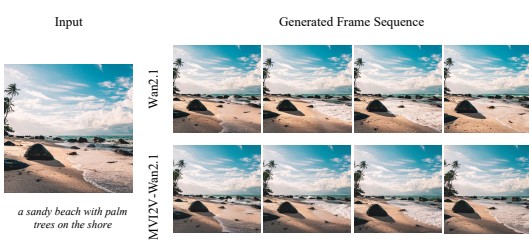

Figure 6: Image-to-Video generation results of our MVI2V-Wan2.1 model.

## 6 CONCLUSION

In this paper, we introduce the task of multi-view guided image-to-video (I2V) generation and propose a comprehensive framework to train a competent model. Starting from a pretrained I2V model, we extend its architecture and incorporate an inpainting-based training strategy to facilitate efficient learning. We also devise an effective data curation pipeline tailored for this task. Given reference images of a person or a garment from various viewpoints, our model can generate dynamic person videos of the subject with a consistent multi-view appearance.

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

## A  THE USE OF LARGE LANGUAGE MODELS(LLMs)

Large language models were employed in this work for the exclusive purpose of stylistic and grammatical enhancement of the manuscript. Their function was confined to improving readability and flow; they were not utilized for any substantive research tasks, including the conception of ideas, design of methodology, analysis of results, or selection of citations. All AI-generated text underwent rigorous review and substantive editing by the authors, who authored and verified all scientific claims, algorithms, and findings presented herein.

## B  MVI2V FOR IN-HOUSE I2V

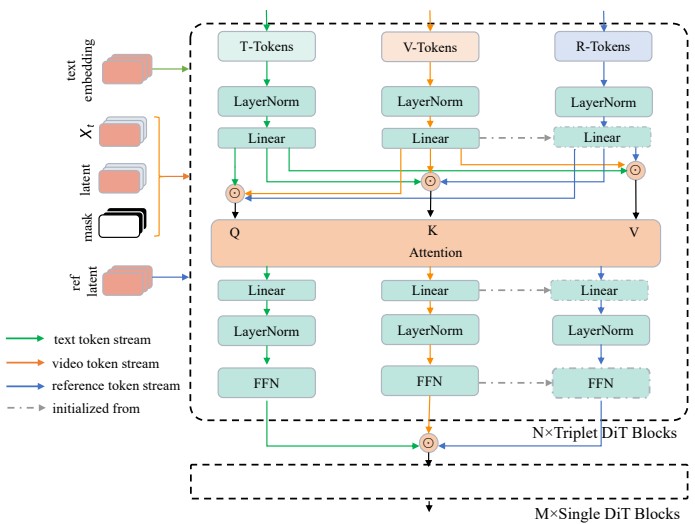

Figure 7: In-house I2V model architecture incorporating MVI2V.

We incorporated the MVI2V method into our in-house mage-to-video generation model by transforming its dual-stream design into a tri-stream architecture. This was achieved by adding a reference token stream, whose modules were initialized with weights from the video token stream. The resulting structure is illustrated in Figure 7.

## C    IMPLEMENTATION DETAILS

**Training Details.** During each training iteration, we randomly sample three reference frames and apply the inpainting sub-task with a 20% probability. For the newly added LoRA layers, the scaling factor $\alpha$ is set to 1 and the lora rank is set to 64. We train the model for 2000 steps using the Adam optimizer ($\beta_1 = 0.9, \beta_2 = 0.999$) Kingma & Ba (2014) with a learning rate of 1e-5 and a batch size of 64. All video frames are resized to a resolution of 1280x720 (720P), while all reference images are resized to 720x720.

**Dataset.** We build our multi-view training dataset from an in-house collection of 1.23M e-commerce garment videos, following the process shown in Fig 3. This initial pool was first filtered to retain only single-person videos, reducing the count to 830K. Subsequently, we applied the large-angle rotation filter, which further narrowed the dataset down to 230K videos. From each of these remaining videos, we extract nine types of reference images as described in 4.

## D    METRICS

**Garment Consistency.** We assess the garment consistency between the generated video frames and the reference images using two distinct models, *i.e.*, a Vision-Language Model (VLM) and DINO. For the VLM-based metric, we adopt the VQAScore methodology, which calculates the generation probability of a target answer. Specifically, we compute the probability of the answer "YES" by applying the softmax function to the logits of the top candidate tokens returned by the model. Let $L_{\text{YES}}$ be the logit for the token "YES" and $T_{\text{candidates}}$ be the set of top candidate tokens, the score is formulated as:

$$p(\text{"YES"}) = \frac{\exp(L_{\text{YES}})}{\sum_{t_i \in T_{\text{candidates}}} \exp(L_{t_i})} \tag{1}$$

If "YES" is not among the top candidate tokens, its probability is taken as 0. We provide detailed instructions for this task in Listing 1. To ensure fairness, we re-grade each video 5 times and take the average probability as the final score ($GPT_{garment}$). For the DINO-based metric, we compute the DINO feature similarity between each generated frame and the reference image. The maximum similarity score across all frames is then taken as the final garment consistency score for the video. We apply the two scoring strategies to the multi-view person and garment evaluation sets, respectively, yields a total of four distinct metrics $GPT_{person}, GPT_{garment}, DINO_{person}, DINO_{garment}$.

**Video Quality.** Consistent with previous work, we evaluate video quality with VBench Zheng et al. (2025). VBench introduces a set of metrics that comprehensively evaluate the videos from both quality and semantic perspectives. We use the "Subject Consistency", "Background Consistency", and "Motion Smoothness" from VBench to evaluate how the two major challenges – temporal consistency and smoothness – are addressed.

**Text-Motion Alignment.** Similar to the VLM-based metric for the garment consistency, we use GPT to evaluate text-motion alignment. For each video, We provide detailed instructions (see Listing 2) to guide the GPT to judge whether the depicted action is consistent with the motion described in the text prompt.

```
# Role
You are an expert visual inspector AI.

# Task
Your task is to determine if the clothing worn by the person in the provided video is an exact match to the
    clothing shown in the provided reference images. You should compare carefully at the color and texture.

# Inputs
- A set of reference images showing a garment from multiple views (front, back, etc.).
- A single video showing a person wearing a garment.

# Rules for Verification
1.  **Appearance Match:** The clothing in the video (including its color, pattern, logos, and design) must be
        identical to the clothing in the reference images.
2.  **Orientation Match:** The garment must be worn correctly. The side identified as the front in the
        reference images must be on the person's front in the video, and the back must be on the person's back.

# Output Instruction
- If AND ONLY IF both rules are strictly met, respond with a single word: YES
- Otherwise, respond with a single word: NO
- Do not provide any explanation, context, or any other text.
```

Listing 1: Instructions for the VLM-based garment consistency evaluation.

```
# Role
You are an expert AI video-text alignment analyst.

# Task
Your task is to determine if the actions performed by the person in the provided video generally align with
    the description in the provided text.

# Inputs
- A text prompt used to generate a video.
- A video generated from the text prompt.

# Rules for Verification
1.  **General Action Match:** The core actions described in the text must be recognizably present in the video
        . An exact, one-to-one match is not required, but the overall action should be similar.
2.  **Consistency:** The actions in the video must not fundamentally contradict the description in the text.

# Output Instruction
- If AND ONLY IF all rules are met, respond with a single word: YES
- Otherwise, respond with a single word: NO
- Do not provide any explanation, context, or any other text.
```

Listing 2: Instructions for the VLM-based text-motion alignment evaluation.

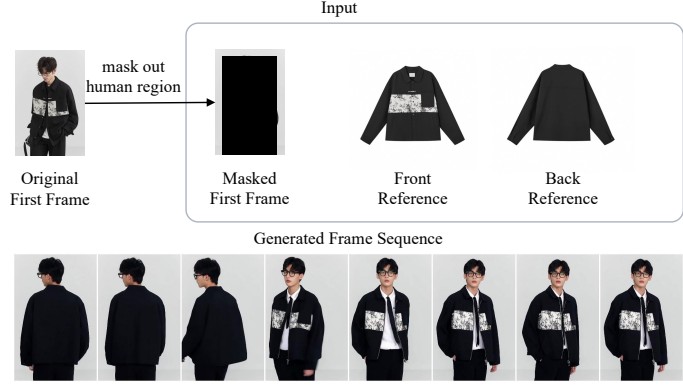

Figure 8: An example of generated results for the inpainting sub-task.

# E  EFFECTIVENESS OF THE INPAINTING SUBTASK

To evaluate the effectiveness of the inpainting sub-task introduced during training, we conducted inference experiments where the human region in the conditioning first frame was masked. The qualitative results are presented in Figure 8. As the figure illustrates, even though the subject's appearance is completely obscured in the first frame, our model successfully reconstructs the appearance of the garment by leveraging the provided reference images. This result validates that

the inpainting sub-task is effective in forcing the model to rely on reference views for appearance synthesis, decoupling it from the conditioning frame.

## F  MORE QUALITATIVE RESULTS

This section provides additional qualitative comparisons against the Wan2.1 I2V model. Figure 9 showcases results generated from garment reference images, while Figure 10 demonstrates performance using person reference images.

810
811
812
813
814
815
816
817
818
819
820
821
822
823
824
825
826
827
828
829
830
831
832
833
834
835
836
837
838
839
840
841
842
843
844
845
846
847
848
849
850
851
852
853
854
855
856
857
858
859
860
861
862
863

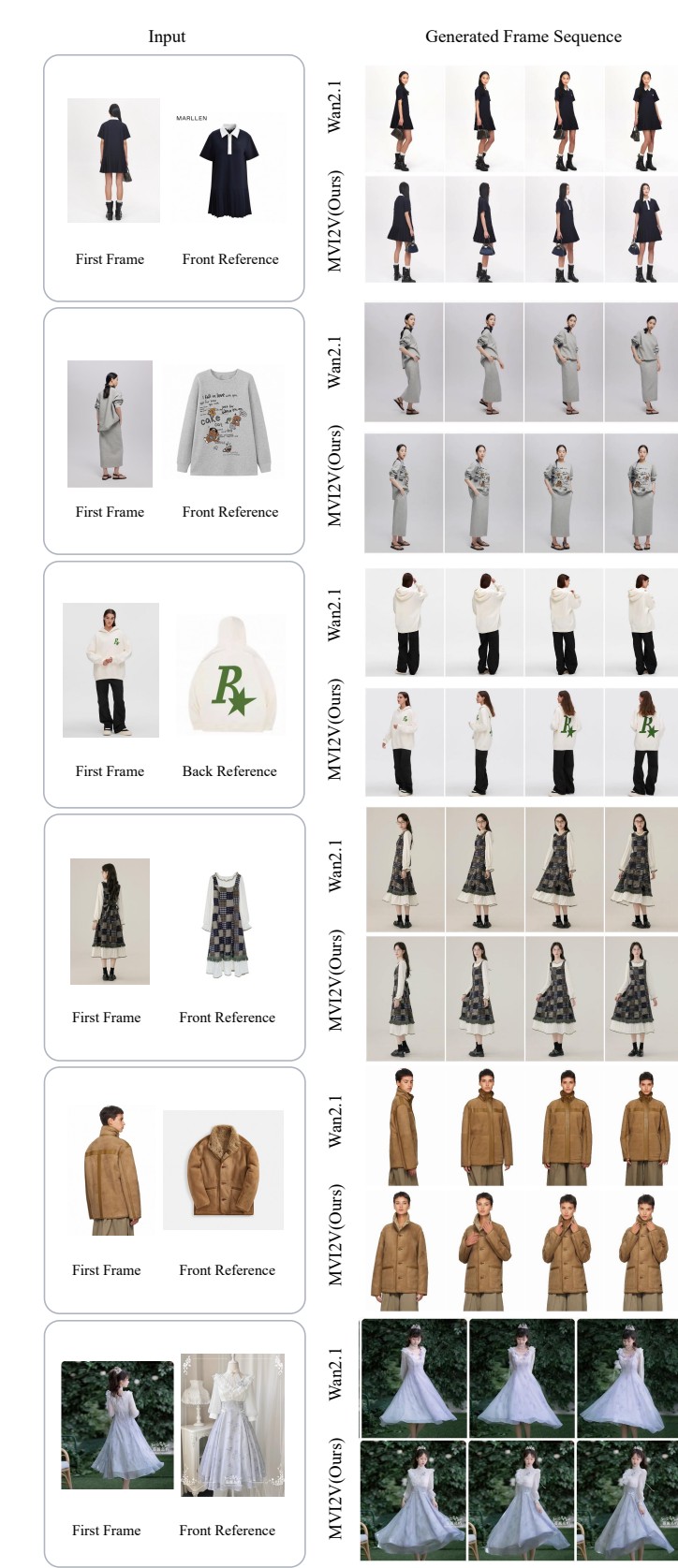

Figure 9: Results generated using garment reference images.

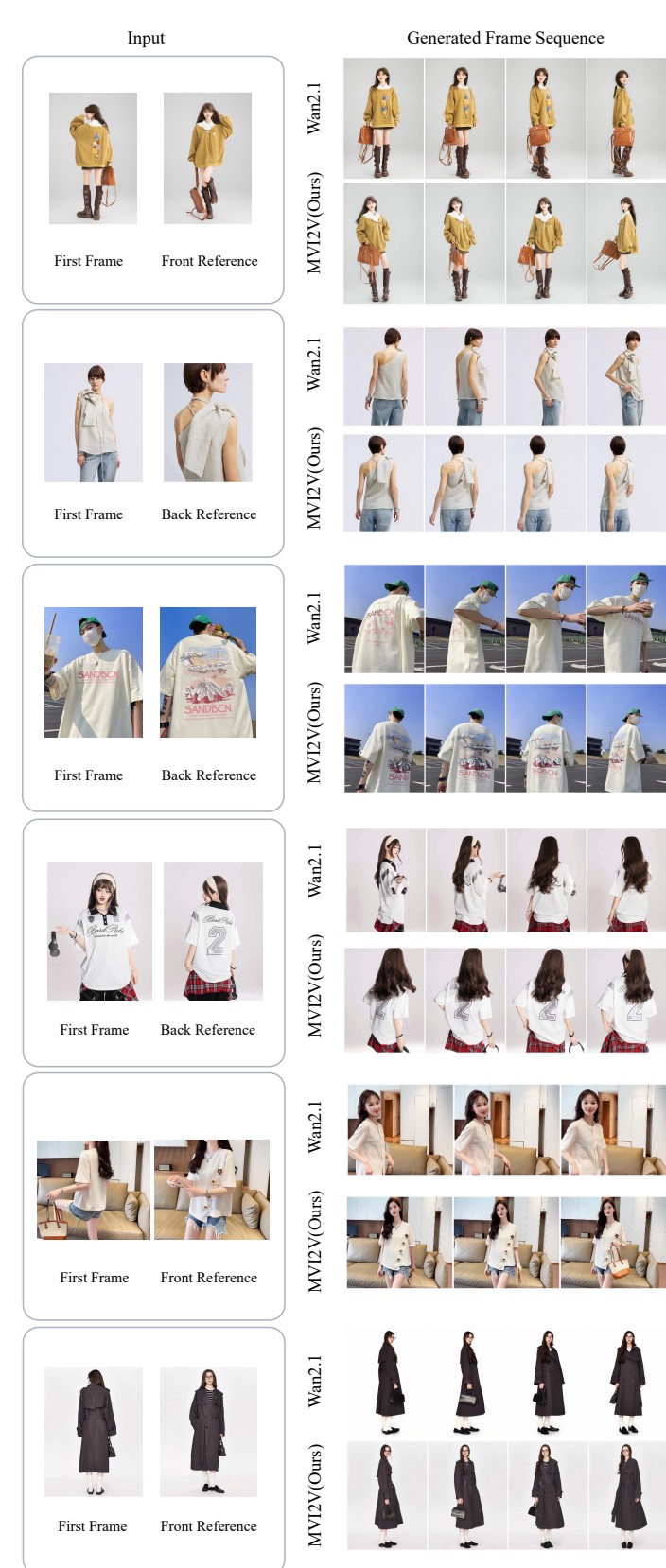

Figure 10: Results generated using person reference images.

