# OpenReview forum: "MVI2V: Human Centric Image to Video Generation with Muliview Consistent Appearance"
_ICLR.cc/2026/Conference — ICLR 2026 Conference Withdrawn Submission_

### Official Review · Reviewer_QqEB · 2025-10-27

**Soundness:** 3
**Presentation:** 2
**Contribution:** 2
**Rating:** 2
**Confidence:** 4

**Summary:**

This paper studies garment consistency in single-person video generation. The authors introduce reference images of the person or their garments from different views as additional conditions for video generation. They also collect a dedicated dataset to train a reference-guided video generation model. The proposed approach achieves better results than the original model without reference guidance.

**Strengths:**

1. The pipeline of building a specific dataset for a targeted task is inspiring.
2. Masking the conditioning image to strengthen the effect of the reference image is a creative idea.

**Weaknesses:**

1. Adding extra conditions to pre-trained image or video generation models is not a new idea. However, the paper does not discuss related works or compare with them experimentally. A particularly relevant work is EasyControl: Adding Efficient and Flexible Control for Diffusion Transformer (ICCV 2025).
2. Although the framework is straightforward, many details are unclear:
    - In Figure 2, it seems that the blue modules are frozen while the orange modules are trained, but this is not stated.
    - In Section 4.3, regarding the diverse reference frame selection, it would help to show the total number of orientation angles and explain why k = 3 is chosen.
    - In the clothing segmentation part of Section 4.3, the definition of the “flattened map” is missing, and the reason for the data scarcity is not discussed.
    - The dataset strategy described in Table 2 is not clearly defined.

    In contrast, Section 4.2 mostly describes standard concepts that are not new and could be shortened to a single paragraph, except for the inpainting part.
3. The experiments are not very convincing:
    - Only pre-trained video generation models are compared, which naturally lack reference-guided capabilities.
    - The results of Wan 2.1 and In-House are inconsistent. It is unclear why text-motion alignment improves significantly in Wan 2.1 but worsens in In-House.
    - The definition of the GPT-related scores is vague. Specifically, how are the top candidate tokens obtained from GPT?
    - The ablation study is incomplete. For instance, why are clothing segments necessary? How does the number of clustered multi-view images affect performance?
4. Equations are not numbered.

**Questions:**

In addition to the issues listed above:
1. Why are different sizes used for the video frames and reference images?
2. What text prompts are used to generate the videos?

---

### Official Review · Reviewer_MZi4 · 2025-10-30

**Soundness:** 3
**Presentation:** 3
**Contribution:** 2
**Rating:** 4
**Confidence:** 4

**Summary:**

This paper focuses on the image-to-video task with multi-view reference images, primarily in e-commerce scenarios. The authors observe that when garment or person references from additional viewpoints are missing, the generated video’s appearance can deviate from reality in certain views. To address this, they introduce multi-view reference images. They adopt a multi-branch DiT framework, adding a new branch to model the reference images and using attention to transmit information. In addition, they propose a data filtering pipeline to curate suitable training data, and an inpainting subtask that compels the model to draw cues from the reference images. Compared with the baselines, the proposed model achieves good results, and the ablation studies validate the rationale behind each design choice.

**Strengths:**

1. This work focuses on the multi-reference image I2V task in the e-commerce domain, where the generated video must maintain appearance consistency with the real person and garments. Without reference information from other viewpoints, the model is forced to imagine content for unseen views. Therefore, the paper introduces multi-view reference images, which achieve good results in experiments.
2. The proposed data construction strategy is reasonable; the introduced inpainting subtask also makes sense and compels the model to learn from the reference images.
3. While common MMDiT architectures duplicate a branch and thus double the parameter count, the authors use LoRA to remain parameter-efficient and improve computational efficiency.
4. The paper is clearly written.

**Weaknesses:**

1. The paper lacks an experiment analyzing how the number of reference images affects the results. I believe this is necessary because, in real applications, sufficient multi-view references may not be available.
2. Could you conduct a comparative experiment using person images versus garment images as references, to observe how each choice influences the results and what different information the model learns from these two types of references.

**Questions:**

1. This work targets a relatively narrow application—primarily model clothing display videos for e-commerce. In my view, for this task a model mainly needs to condition on the first-frame image and incorporate clothing or person information from other viewpoints, so the problem does not seem difficult. I would be happy to hear the authors’ perspective on where the core challenges actually lie.

2. The paper focuses on a multi-branch DiT architecture that can be adapted to single-stream or dual-stream video models. A new branch is introduced to process reference images, and attention is used to transmit information. The experiments compare this with a token-concat approach, which I understand simply concatenates tokens from the video and the reference images; the ablations show MMDiT performs better. That said, I still have reservations about the necessity of MMDiT, because token-concat (a context-frame method) is more implementation-efficient and requires no architectural changes. In addition, some recent work has obtained strong results with token-concat (e.g., FullDiT [1]), albeit on control-oriented tasks. I would like to know, under the same experimental setting, why token-concat performs poorly on this task (the table suggests a sizable gap in the metrics).

[1] X. Ju, et al., “FullDiT: Multi-task Video Generative Foundation Model with Full Attention,” arXiv, arXiv:2503.19907, 2025.

---

### Official Review · Reviewer_x69t · 2025-10-31

**Soundness:** 3
**Presentation:** 3
**Contribution:** 2
**Rating:** 4
**Confidence:** 4

**Summary:**

This paper presents MVI2V, a practical framework for multi-view–enhanced image-to-video (I2V) generation. The method augments pretrained I2V diffusion backbones with a dedicated reference-image stream, a lightweight inpainting subtask that encourages effective use of reference information, and a carefully designed multi-view training pipeline. Implemented as a parameter-efficient, LoRA-style extension to transformer blocks, MVI2V is evaluated on both (1) a newly curated small-scale multi-view benchmark and (2) standard I2V benchmarks. Experimental results demonstrate good performance in garment consistency while largely preserving the baseline I2V model’s general generation capabilities.

**Strengths:**

1. The overall organization of this paper is good, which is easy to understand.
2. The triplet/dual-stream extension, combined with the LoRA-style parameter-efficient adaptation, can be seamlessly integrated into various diffusion backbones, enhancing the framework’s general applicability.

**Weaknesses:**

1. The paper tackles the problem of visual inconsistencies in unseen regions for single-image-to-video (I2V) generation by leveraging multi-view imagery. While this is a practical and well-established strategy, the core contribution is deemed incremental. The proposed architecture demonstrates a straightforward application of existing concepts, primarily extending a pre-trained I2V model with additional branches to process reference images. Consequently, the work appears to fall more accurately into the domain of multi-image-to-video generation, which inherently simplifies the consistency challenge, rather than introducing a novel solution for the more constrained single-image setting.

2. The implementation details provided are insufficient for a full understanding and reproduction of the work. Several critical components are inadequately described. Specifically, the encoding process for multiple reference images into the unified feature C_ref, and the subsequent processing of the derived feature y_3, remain unclear. Furthermore, while the first frame is encoded with CLIP, the rationale for omitting any masking strategy is not explained. Figure 2 suggests that the Wan Encoder produces at least four distinct features, yet their respective roles and processing flows are not elaborated. The method for acquiring the front and back view images, as mentioned in the "Diverse Reference Frames Selection" section, is unspecified. Finally, essential experimental details such as the loss function, hardware configuration, and training software stack are omitted, which are necessary to assess the experimental setup.

3. The experimental evaluation raises concerns regarding the fairness of comparisons and the depth of analysis. The comparison against Wan2.1 may not be entirely equitable, as it is a single-image method and does not utilize the additional multi-view information available to the proposed model. A more compelling demonstration of efficacy would involve comparisons against other multi-image input approaches, or the establishment of a corresponding baseline that incorporates multi-view data. Furthermore, the paper lacks a thorough analysis of the performance degradation in Text-Motion Alignment observed for the MVI2V-In-House method in Table 1. Finally, an analysis of computational complexity and inference time, critical for assessing the model's practical efficiency, is absent.

Minor weaknesses:
1. Ablation studies on key parameters (e.g., the number of reference images and inpainting probability) are needed to understand their sensitivity.
2. A systematic analysis of failure cases is absent.
3. The format of references is inconsistent.

**Questions:**

1. Please clarify how multiple reference images are encoded and fused into a unified representation? Are they processed independently and then pooled, or is there a more sophisticated fusion mechanism?

2. What is the relative increase in parameter count and inference time due to the added reference stream and LoRA layers? Is the method suitable for real-time or near-real-time applications?

3. How does the model performance change with fewer reference images? Is there a minimum set of views required for robust performance?

---

### Official Review · Reviewer_C1Fg · 2025-11-03

**Soundness:** 2
**Presentation:** 2
**Contribution:** 2
**Rating:** 4
**Confidence:** 4

**Summary:**

This paper proposes MVI2V, a framework for enhancing image-to-video generation with multi-view reference images to improve garment and appearance consistency. The method extends existing I2V models with an additional forward stream for reference images, introduces an inpainting subtask to encourage reliance on references, and designs a data curation pipeline for training. Experiments on Wan2.1 and an in-house model show improvements in garment consistency metrics.

**Strengths:**

The task of multi-view consistent I2V generation is novel and practically relevant, especially for e-commerce applications.

The proposed architecture is modular and compatible with both single-stream and dual-stream backbones.

The data curation pipeline and inpainting subtask are well-motivated and empirically validated.

The paper includes extensive experiments and ablations, demonstrating the contribution of each component.

**Weaknesses:**

Limited novelty in core technique: The architectural extension (adding a stream for reference images) is a straightforward adaptation of MMDiT-style fusion, and the inpainting strategy, while effective, is not highly innovative.

Evaluation scope: The proposed benchmark is small (205 samples) and internally constructed, lacking comparison to public benchmarks or state-of-the-art multi-view or garment-aware generation methods.

**Questions:**

No Questions

---

### Note · Authors · 2025-11-13

I have read and agree with the venue's withdrawal policy on behalf of myself and my co-authors.